# Effects of Problem-Based Learning on the Problem-Solving Ability and Self-Efficacy of Students Majoring in Dental Hygiene

**DOI:** 10.3390/ijerph19127491

**Published:** 2022-06-18

**Authors:** Jin-Sun Choi, Soo-Myoung Bae, Sun-Jung Shin, Bo-Mi Shin, Hyo-Jin Lee

**Affiliations:** 1Department of Dental Hygiene, College of Dentistry, Gangneung-Wonju National University, Gangneung-si 25457, Korea; jjcjs@gwnu.ac.kr (J.-S.C.); edelweiss@gwnu.ac.kr (S.-M.B.); freshjung@gwnu.ac.kr (S.-J.S.); purplebom@gwnu.ac.kr (B.-M.S.); 2Research Institute of Oral Science, Gangneung-Wonju National University, Gangneung-si 25457, Korea

**Keywords:** dental hygiene, problem-based learning, problem-solving ability, self-efficacy

## Abstract

This study developed a problem-based learning (PBL) module to improve integrated thinking and problem-solving ability in students of dental hygiene. After applying PBL, the study tested the improvement in the problem-solving ability and self-efficacy of students. The subjects were 31 fourth-year students of the Department of Dental Hygiene at G University. The PBL process was applied to three topics for 15 weeks, and the tools for evaluating problem-solving ability and self-efficacy were reconstructed and used before and after the application to examine the effects of the module. The result indicates that the mean of problem-solving ability (32 detailed items) increased from 3.37 to 3.65 (an increase of 0.28) after classes (*p* < 0.001). Alternatively, the average for self-efficacy (22 detailed questions) increased from 0.21 to 2.89 (*p* < 0.05; an increase of 2.67). The study also confirmed the correlation between problem-solving ability and the total posttest score for self-efficacy (*p* < 0.001). Thus, the problem-solving ability and self-efficacy of learners were improved in the class to which PBL was applied. These improvements exerted a significant effect on the improvement of problem-solving ability. This finding confirmed the effect of the PBL method on dental hygiene education.

## 1. Introduction

A dental hygienist is a licensed health care professional and a clinician with the overall role of providing preventive, educational, and therapeutic services for the management of oral diseases and the promotion of oral health [1]. Accordingly, training dental hygienists to acquire the expertise in coping with various practical situations that may occur in clinical settings is extremely important. In this regard, dental hygiene education strives to train clinicians who can effectively solve problems in various clinical situations by utilizing knowledge and experience. Additionally, as an alternative to increasing the ability to cope with the abovementioned situations, endeavors have been made to provide education that applies problem-based learning (PBL) [2].

PBL was first introduced at McMaster University Medical School in Canada during the late 1960s [3]. Medical schools in Korea also implemented the PBL process in 87.5% (35 out of 40) of schools, where the majority of schools have been applying PBL [4]. Education applying the PBL has been tried not only in the medical field but also in other fields, such as nursing, dentistry, and dental hygiene [5,6,7]. The reason for which PBL is attracting attention in medical and health academia is that it is evaluated to improve students’ ability to apply knowledge to clinical sites. That is to say, this learning method applies knowledge and technology to solve problems using clinical data [8,9].

Problem-solving ability is a major competency for solving practical problems in clinical situations and is a requirement for the professional performance of the role of a dental hygienist. Additionally, problem-solving skills are essential for dental hygienists in instructing, collecting, analyzing, and evaluating accurate understanding and information on a given situation in the clinical dental field. Additionally, self-efficacy is a competency necessary for dental hygienists in acting as an important variable in improving job satisfaction.

Relevant studies reported that the PBL process improved the self-efficacy and problem-solving ability of students in medicine, dentistry, and nursing [10,11,12,13]. Other studies also reported that self-efficacy exerted an influence on job commitment and major satisfaction [14,15], whereas problem-solving ability helped improve job performance [16]. Given these results, educational methods, such as PBL, should be reflected to improve the self-efficacy and problem-solving ability of Dental Hygiene students. However, studies that developed the PBL process in the field of dental hygiene and evaluated its effects, such as the enhancement of self-efficacy and problem-solving ability, are very few, especially in Korea.

Thus, this study aimed to develop a PBL module to improve the problem-solving ability of students through integrative thinking and self-efficacy in dental hygiene, and then to apply it. The study was conducted for one semester to evaluate the improvement in problem-solving ability and self-efficacy among students before and after PBL in dental hygiene.

## 2. Materials and Methods

### 2.1. Participants

This study was conducted with research ethics approval from the Institutional Review Board of G University (IRB No: GWNUIRB-2020-25). The subjects were 31 students enrolled in a course called Clinical Dental Hygiene 3 (oral health management for special patients) in the second semester of the fourth year (the last semester of an undergraduate course). Clinical Dental Hygiene 3 is a course for achieving integrated thinking skills based on the basic knowledge of students majoring in clinical dental hygiene. We selected all students who participated in the class and collected the informed consent for this study. We also provided sufficient opportunities to opt out. Clinical Dental Hygiene 3 was conducted for one semester from September to December 2020. The levels of problem-solving ability and self-efficacy were measured once at start of the class in the first week and once at end of the class in the 15th week, respectively. The researcher (a professor for this class) provided sufficient information on this course to students and conducted the questionnaire to students who agreed in writing.

In this study, the theme of the PBL module was composed of three main problems, namely, the elderly, orthodontics, and dental implants, which were discussed by four researchers with reference to oral health-related issues and clinical cases. Two out of four researchers operated classes applying PBL modules. In addition, two researchers held a weekly meeting to standardize the content of class guidance. The theme of the PBL module set problematic scenarios that dental hygiene graduates may experience in dental clinics. The students identified themselves with the main character of the problem situation to induce self-directed learning. By constructing a practical situation for students, we induced interest and intrinsic motivation for learning. Additionally, we developed a dental hygiene care process chart and an oral photograph example as reference materials related to the problem situations to ensure that the problem-solving process resembled the actual situation.

In the scenario of the three main problems, the role of students was specified to enable them to take the initiative to solve problems. The study included problem situations that could be approached using various methods to derive various solutions and that were presented as situations that can be solved through team cooperation.

The PBL process was applied to the three problems for 15 weeks. The process is composed of (A) presenting and analyzing problems for each topic, (B) writing a task execution plan and individual learning, (C) searching for solutions to the problem, (D) proposing solutions to the problem and presenting the result, and (E) conducting presentations for 4 weeks

### 2.2. Statistical Analysis

To evaluate the effects of PBL on the problem-solving ability and self-efficacy of students, previously developed tools for assessing problem-solving ability and self-efficacy before and after the PBL class were reconstructed and used [17]. Problem-solving ability was evaluated on a scale of 1–5, whereas self-efficacy was evaluated on a scale of 1–4. The levels of problem-solving ability and self-efficacy were evaluated once each before and after the application of the PBL module. A survey was conducted on a total of 31 students, and the survey response rate was confirmed to be 100%.

Problem-solving ability comprised 32 items, whereas the mean reliability (coefficient α) for the questionnaire items was 0.88. A negative question was calculated using inverse calculation. Moreover, self-efficacy comprised 22 items with coefficient α of 0.92 for the questionnaire items.

Data were computerized and analyzed using SPSS 25.0 (SPSS Inc., Chicago, IL, USA). For analysis, normality was tested using the Kolmogorov–Smirnov test followed by a paired-sample *t*-test. The data were normal and Spearman’s correlation analysis was performed to confirm the correlation between problem-solving ability and self-efficacy.

## 3. Results

The mean of problem-solving ability improved by 0.28 (from 3.36 to 3.64), which was statistically significant (*p* < 0.001).

In the results of items 4, 10, 17, 18, 22, 24, and 31 (Table 1), the score increased after the PBL from 0.23 min to 0.84 max and was statistically significant (*p* < 0.05). 

Alternatively, items 16 and 27 decreased by 0.1 and 0.29, respectively, after the PBL and were statistically nonsignificant.

The mean of self-efficacy improved by 0.21 (from 2.67 to 2.89), which was statistically significant (*p* < 0.001). Item 4 decreased by 0.13 but was statistically nonsignificant (Table 2). For all other questions, the results demonstrated improvements ranging from 0.10 to 0.52, especially for items 1, 6, 9, 13, 14, 15, and 20, which were statistically significant (*p* < 0.05).

The correlation between the subject’s problem-solving ability and the posttest total score of self-efficacy was checked. Spearman’s rank correlation analysis between problem-solving ability and self-efficacy was 0.626, which indicates a significant correlation and demonstrates a positive (+) correlation and a statistically significant difference (*p* < 0.001) (Table 3). In other words, improvement in self-efficacy exerted a significant effect on the improvement in problem-solving ability.

## 4. Discussion

Dental hygienists may face several problems related to the oral health of dental patients and should solve problems based on critical thinking and decisions. In this study, PBL class was applied to students who major in dental hygiene to improve not only problem-solving ability but also self-efficacy.

This study found that the levels of problem-solving ability and self-efficacy increased after applying the PBL class. These findings were similar to those of previous studies that focused on the effects of PBL class in dentistry, nursing, and medicine [18,19,20,21].

According to previous studies that evaluated the effect of PBL on problem-solving abilities, Choi et al. [22] revealed that the learning outcomes of PBL improved across abilities of problem solving, self-directed learning, and critical thinking compared with traditional lectures in nursing education. In Korea, Lee et al. [21] reported that PBL in the field of dentistry improved self-directed learning, communication ability, and problem-solving ability. In particular, the study confirmed that the problem-solving ability could exert a positive effect for one year (up to the second semester) after applying PBL.

Problem-solving ability is defined as a skill in identifying a problem and taking action to solve it [23]. This ability is crucial among dental professionals, including dental hygienists, in identifying and solving the problems of patients. If health professionals possess high levels of problem-solving ability, then they can more effectively analyze the health problems of patients. Finally, professionals could determine solutions that target the root causes of health problems and perform intervention plans [24,25]. Dental hygienists may be the first to face various patients in dental clinics and be required to identify the problems of patients well. Hence, dental hygienists must possess the problem-solving ability to identify and resolve patient problems clearly. To develop these abilities, the process of PBL can be a good method for learning through hypothetical and actual clinical cases.

PBL, in which students undergo a process of solving problems within a team based on actual clinical problems, operates in an entirely different manner from traditional lecture classes [26]. Scholars reported learning effects as improvements in problem-solving ability, creativity, and self-directed learning [18,22]. In this process, students learn and solve problems based on knowledge and experience with a focus on the subject instead of the disease. Moreover, the experience is objectified through discussion and reflection, and knowledge is transferred [27].

The present study found that the self-efficacy of students was also improved after applying the PBL class. Choi and Kim [20] reported that students majoring in medicine exhibited high levels of self-efficacy and satisfaction after participating in PBL classes. Additionally, they confirmed that self-efficacy had increased over time.

Self-efficacy is defined as judgments students make about their ability in specific situations [28] and exerts an influence on not only decisions related to learning and knowledge maintenance but also planning and implementing learning activities [29]. It can also be an important factor that positively influences motivation, attitude, and learning outcomes [30]. If self-efficacy is high, then it can control learning and improve self-control. Kokcu et al. [31] reported the higher the self-efficacy, the higher the problem-solving ability. Thus, self-efficacy, as well as problem-solving ability, could be an important aspect to be considered among students majoring in dental hygiene. The self-efficacy of students of dental hygiene should be improved through the process of self-directed learning such as PBL. Based on the results of the present study, PBL could be an effective strategy for enhancing the problem-solving ability and self-efficacy of students in dental hygiene.

One thing to note is that the level of problem-solving ability and self-efficacy were significantly improved in the overall score, however, some specific items were not significantly improved. Considering only the overall score of improvement in problem-solving ability and self-efficacy, it might be mistaken for a comprehensively significant improvement.

This study has its limitations. It aims to evaluate the effect of PBL at a university, which may be difficult to judge as a general result in dental hygiene. Additionally, other characteristics related to problem-solving ability and self-efficacy may have been insufficiently examined. Because there were no comparisons between the PBL and other educational intervention, the effects of only PBL on the problem-solving ability and self-efficacy should be evaluated through the study design for controlling and adjusting other interventions in further studies. This study explored this topic with certain constraints because the learning process of the students was applied through a combination of face-to-face and remote learning due to restrictions related to the coronavirus disease 2019 pandemic. In this study, the main results mean there is a short-term effectiveness of PBL on problem-solving ability and self-efficacy. In order to evaluate the long-term effectiveness, we should design the longitudinal studies and evaluate the outcomes as time goes by.

Nonetheless, the study remains meaningful because it examined the effects of PBL on education related to dental hygiene, which is an aspect that lacks scholarly attention.

## 5. Conclusions

Classes to which PBL was applied improved the problem-solving ability and self-efficiency of learners. Additionally, the improvement in self-efficacy exerted a substantial influence on the improvement of problem-solving ability. These results confirmed that the educational effect of PBL classes could be verified and that education using the PBL method could be applied to medical courses, such as dental hygiene. In the future, various PBL problem scenarios based on actual clinical settings and PBL modules that can be applied by grade should be developed. Through this, a long-term course should be established to develop integrated thinking and problem-solving skills by applying PBL module classes to all grades.

## Figures and Tables

**Table 1 ijerph-19-07491-t001:** Problem-solving ability of subjects.

Variable		Before PBL	After PBL	*p*-Value ****
		Mean ± SD	Mean ± SD	
	Total	3.36 ± 0.41	3.64 ± 0.49	<0.001
1. *	I am going to find out why something does not work out when it does not.	4.03 ± 0.75	4.06 ± 0.96	0.831
2. *	I do not bother gathering information to figure out how to solve something complicated.	3.35 ± 0.98	3.58 ± 1.03	0.109
3. *	If I cannot solve a problem at once, I get anxious because it seems that I do not have the ability to solve the problem.	2.84 ± 1.07	3.35 ± 1.14	0.033
4. *	After I fix something, I do not look at what went right and what went wrong.	3.26 ± 0.96	3.97 ± 0.71	<0.001
5.	I can usually come up with a number of creative and effective ways to solve any problem.	3.16 ± 0.93	3.45 ± 0.72	0.119
6.	After trying to solve a problem, I take the time to compare the actual result with what I expected.	3.16 ± 1.07	3.45 ± 0.81	0.248
7.	When I have a problem, I think about solutions a lot until I cannot come up with any more ideas.	2.94 ± 1.15	3.23 ± 1.02	0.130
8.	I constantly look for changes in my feelings when I encounter a certain problematic situation.	3.42 ± 0.92	3.81 ± 0.95	0.070
9.	I have the ability to solve most problems, even if the problem is so difficult that at first there seems no right solution.	3.35 ± 0.80	3.61 ± 0.84	0.058
10. *	A lot of the things I run into are too complex for me to deal with.	3.23 ± 0.84	3.61 ± 1.05	<0.05
11. *	When I make a decision on a problem, I am satisfied with that decision even after that.	3.23 ± 0.84	3.26 ± 1.09	0.851
12. *	When I run into something, the first thought that comes to my mind is to solve it.	2.84 ± 0.97	2.97 ± 1.05	0.423
13. *	Sometimes I mess things up by rushing things rather than taking the time and step-by-step solve them.	3.55 ± 0.93	3.55 ± 0.99	1.000
14. *	When deciding how to solve a problem, I do not weigh every single probability of success.	3.61 ± 0.88	3.77 ± 0.72	0.344
15.	When I run into a problem, I think first, and then decide what to do next.	4.06 ± 0.63	4.16 ± 0.58	0.414
16. *	I do things according to the first thought that comes to mind.	3.06 ± 1.00	2.97 ± 0.95	0.557
17.	When I make a decision, I think about the results of how to solve the problem and compare them one by one.	3.29 ± 0.86	3.84 ± 0.78	<0.05
18.	When I make plans to solve problems, I am confident that I can put those plans into action.	3.16 ± 0.93	3.61 ± 0.80	<0.05
19.	I try to know in advance how the actions I take will affect the whole.	3.74 ± 0.73	3.84 ± 0.64	0.522
20. *	I think of only one way to solve a problem and no other way.	3.58 ± 0.92	3.87 ± 0.81	0.071
21.	I believe that most problems that happen to me can be solved if you put in the time and effort.	3.84 ± 0.73	4.03 ± 0.75	0.226
22.	I have the confidence to handle any problematic situation that I have never experienced before.	3.00 ± 1.00	3.55 ± 0.85	<0.05
23. *	Even when you start taking action to solve a problem, you sometimes get lost or distracted.	3.00 ± 0.89	3.23 ± 0.88	0.214
24. *	I quickly make a decision and soon regret it.	3.06 ± 0.93	3.90 ± 0.83	<0.001
25.	I believe I have the ability to solve even the most difficult problems I see for the first time.	3.00 ± 0.77	3.65 ± 0.88	0.001
26.	I compare various problem-solving methods and make decisions.	3.55 ± 0.81	3.77 ± 0.72	0.182
27. *	When I run into a problem, I do not review what things around me will help solve the problem.	4.16 ± 0.58	3.87 ± 0.92	0.107
28	When I am confused about a problem, the first thing I do is identify the problem and come up with all the relevant information.	3.32 ± 0.83	3.81 ± 0.75	<0.05
29 *	Sometimes I get so caught up in my emotions that I cannot think of different ways to deal with the problem.	3.35 ± 0.98	3.71 ± 0.97	0.009
30	I think that the actual results of the decisions I have made are generally similar to the results I expected.	3.32 ± 0.65	3.45 ± 0.72	0.325
31 *	I am not sure if I can solve any problems if they happen.	3.39 ± 0.76	3.74 ± 0.73	<0.05
32	When I have a problem, I have the ability to figure out exactly what the problem is.	3.94 ± 0.51	4.10 ± 0.79	0.169

* Negative questions, ** *p*-value by paired *t*-test.

**Table 2 ijerph-19-07491-t002:** Self-efficacy of subjects.

Variable		Before the PBL	After the PBL	*p*-Value ***
		Mean ± SD	Mean ± SD	
Total		2.67 ± 0.38	2.88 ± 0.34	<0.05
1.	I think most of the content can be learned well when you start studying at school.	2.94 ± 0.44	3.13 ± 0.43	<0.05
2.	Even if the content is complex, I keep trying until I understand it.	2.94 ± 0.63	3.29 ± 0.53	0.001
3.	When I study, I never stop until I have done as much as I planned.	2.32 ± 0.70	2.52 ± 0.81	0.226
4.	When I study, if something difficult comes up, I always understand and move on.	2.94 ± 0.57	2.81 ± 0.75	0.325
5.	When I study, I keep working hard until I achieve the goals I set.	2.74 ± 0.58	2.87 ± 0.56	0.211
6.	I am confident that I can do well even if the content of learning is difficult.	2.61 ± 0.62	2.94 ± 0.57	<0.05
7.	When I study, I am confident in most content.	2.48 ± 0.68	2.71 ± 0.64	0.147
8.	When I study, even if it is something I do not want to study much, I study it to the end.	2.52 ± 0.72	2.71 ± 0.90	0.206
9.	When I decide to study, I start immediately.	2.19 ± 0.60	2.71 ± 0.90	<0.05
10.	I can continue studying even if something interferes with my study.	2.13 ± 0.72	2.29 ± 0.82	0.305
11.	I keep trying until I understand what I am learning, even if I find it difficult.	2.90 ± 0.60	3.03 ± 0.55	0.211
12.	When I try to learn something new, I never give up, even if I find it difficult at first.	2.74 ± 0.68	2.87 ± 0.72	0.325
13.	I think I will be able to study well in the future.	2.65 ± 0.61	3.13 ± 0.62	<0.001
14.	I think most of my learning methods are effective.	2.55 ± 0.62	2.90 ± 0.60	<0.05
15.	I believe in my own abilities in school studies.	2.68 ± 0.60	2.94 ± 0.63	<0.05
16.	When I study, I do not give up easily on most of the content.	2.68 ± 0.60	2.84 ± 0.64	0.134
17.	It is easy to see why I have difficulties when studying.	2.87 ± 0.62	3.03 ± 0.66	0.231
18.	When new learning content comes out, I tend to understand it very quickly.	2.32 ± 0.60	2.58 ± 0.72	0.073
19.	Failure in exams makes me try harder.	2.74 ± 0.82	2.87 ± 0.76	0.380
20.	When it comes to learning a new subject, you usually know how to study.	2.52 ± 0.68	2.77 ± 0.50	<0.05
21.	I can plan and start studying on my own without the help of others.	3.10 ± 0.60	3.23 ± 0.50	0.103
22.	I believe that you can study difficult content well if you put in effort.	3.23 ± 0.50	3.32 ± 0.48	0.414

** p*-value by paired *t*-test.

**Table 3 ijerph-19-07491-t003:** Correlation between problem-solving ability and self-efficacy.

	Problem-Solving Capability	Self-Efficacy
Problem-solving capability	1.000	0.626 **
Self-efficacy	0.626 **	1.000

** *p* < 0.001.

## Data Availability

Not applicable.

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
