# Peer review of "Effects of Problem-Based Learning on the Problem-Solving Ability and Self-Efficacy of Students Majoring in Dental Hygiene"

_ijerph, 2022, doi:10.3390/ijerph19127491_

Round 1
Reviewer 1 Report
The article is an excellent example of a qualitatively conducted study: the methodology is described in sufficient detail to allow the experiment to be repeated; the presentation is logically consistent; simultaneously, the studies are presented briefly, without an artificial increase in volume. Separately, it is worth noting the Discussion section, which fully reflects the requirements for this section. I advise you to publish the article, however, after making certain corrections, namely:
- the authors did not indicate the period for which the study was conducted;
- section Conclusions should be improved: describe the ways of further research, show the main results and highlights of the work.
- It is worth pointing out the limitations of the study.
Author Response
Dear Reviewer 1.
Thank you very much for reviewing my paper. All your comments have contributed to making my paper of a higher quality. I have tried to make most of the changes to the part you suggested. I sincerely hope that the part I have revised will be agreeable with you.
The article is an excellent example of a qualitatively conducted study: the methodology is described in sufficient detail to allow the experiment to be repeated; the presentation is logically consistent; simultaneously, the studies are presented briefly, without an artificial increase in volume. Separately, it is worth noting the Discussion section, which fully reflects the requirements for this section. I advise you to publish the article, however, after making certain corrections, namely:
- the authors did not indicate the period for which the study was conducted;
- Thank you so much for your comment. I added the description in the Materials and Methods section according to the reviewer’s suggestion as below.
- Page 2. Line 87-90. 2. Materials and Methods (Red font)
We selected all students who participated in the class and suggested the informed consent for this study. We also informed sufficient opportunities to opt out. Clinical dental hygiene 3 was conducted for one semester from September to December 2020.
- section Conclusions should be improved: describe the ways of further research, show the main results and highlights of the work.
- Thank you so much for your comment. I added the description in the conclusions section according to the reviewer’s suggestion as below.
- Page 8. Line 229-232. 5. Conclusions (Red font)
In the future, various PBL problem scenarios based on actual clinical settings and PBL modules that can be applied by grade should be developed. Through this, a long-term course should be established to develop integrated thinking and problem-solving skills by applying PBL module classes to all grades.
It is worth pointing out the limitations of the study.
- Thank you so much for your comment. I added the limitations in the discussion section according to the reviewer’s suggestion as below.
- Page 8. Line 218-220. 4. Discussion (Red font)
In this study, the main results mean the short-term effectiveness of PBL on problem-solving ability and self-efficacy. In order to evaluate the long-term effectiveness, we should design the longitudinal studies and evaluate the outcomes as time goes by.

Reviewer 2 Report
This article assesses effects of problem-based learning (PBL) on problem-solving ability and self-efficacy among students majoring in dental hygiene. As dental hygiene education aims to train clinicians to effectively problem-solve in various situations, authors describe how efforts have been made to incorporate PBL into dental hygiene education. Authors assert that while studies have reported that PBL improved the problem-solving ability and self-efficacy among students in medicine, dentistry, and nursing, studies that assess effect of PBL on dental hygiene students are very few, especially in Korea. Thus, authors aim to address this gap in the literature with their study.
The review is as follows:
- Regarding Materials and Methods, how was the study advertised to students?
- Who delivered the questionnaire to students?
- At what time points during the semester were the surveys administered?
- Also, authors should clarify the study design. Is it pre-test/posttest? If so, please specify.
- For the PBL, authors should specify who delivered the curriculum.
- Regarding Results, what was the response rate for survey completion?
Overall, this is a unique, pertinent paper on a relevant issue. It is an interesting topic to read about. This study would address a gap in the literature based on the population of focus in the study. The Materials and Methods need further development and more clarity. Authors should clarify the survey administration procedures and timeframes for this study. Tending to these areas may help to improve the paper.
Author Response
Dear Reviewer 2.
Thank you very much for reviewing my paper. All your comments have contributed to making my paper of a higher quality. I have tried to make most of the changes to the part you suggested. I sincerely hope that the part I have revised will be agreeable with you.
This article assesses effects of problem-based learning (PBL) on problem-solving ability and self-efficacy among students majoring in dental hygiene. As dental hygiene education aims to train clinicians to effectively problem-solve in various situations, authors describe how efforts have been made to incorporate PBL into dental hygiene education. Authors assert that while studies have reported that PBL improved the problem-solving ability and self-efficacy among students in medicine, dentistry, and nursing, studies that assess effect of PBL on dental hygiene students are very few, especially in Korea. Thus, authors aim to address this gap in the literature with their study.
The review is as follows:
-Regarding Materials and Methods, how was the study advertised to students?
Who delivered the questionnaire to students?
- Thank you so much for your comment. I have added a part of the Materials and Methods section according to the reviewer’s suggestion.
- Page 2-3. Line 91-93. 2. Materials and Methods (Red font)
The researcher (a professor for this class) provided sufficient information on this course to students and conducted the questionnaire to students who agreed in writing.
-At what time points during the semester were the surveys administered?
- Thank you so much for your comment. I have added a part of the Materials and Methods section according to the reviewer’s suggestion.
- Page 2. Line 89-91. 2. Materials and Methods (Red font)
Clinical dental hygiene 3 was conducted for one semester from September to December 2020. The levels of problem-solving ability and self-efficacy were measured once at start of class in the first week, and once at end of class in the 15th week, respectively.
-Also, authors should clarify the study design. Is it pre-test/posttest?
- Thank you so much for your comment. I have added a part of the Materials and Methods section according to the reviewer’s suggestion.
- Page 3. Line 120-121. 2. Materials and Methods (Red font)
The levels of problem-solving ability and self-efficacy were evaluated once each before and after the application of the PBL module.
-If so, please specify.For the PBL, authors should specify who delivered the curriculum.
- Thank you so much for your comment. I have added a part of the Materials and Methods section according to the reviewer’s suggestion.
- Page 3. Line 96-98. 2. Materials and Methods (Red font)
Two out of four researchers operated classes applying PBL modules. In addition, two researchers held a weekly meeting to standardize the content of class guidance.
-Regarding Results, what was the response rate for survey completion?
- Thank you so much for your comment. I have added a part of the Materials and Methods section according to the reviewer’s suggestion.
- Page 3. Line 121-122. 2. Materials and Methods (Red font)
- A survey was conducted on a total of 31 students, and the survey response rate was confirmed to be 100%.

Reviewer 3 Report
This is a study of a problem-based learning intervention involving 31 fourth-ear dental hygienist students at Gangneung National University. The intervention consisted of 32 detailed items assessing problem-solving ability and 22 detailed questions relating to self-efficacy. The study found significant increases in both measures with an increase of 0.284 problem solving ability and 2.674 self-efficacy.
The research is straightforward and scientifically conducted. That notwithstanding, there are some pretty serious issues that need to be dealt with as described below.
Problem-based learning has been used in a wide variety of settings. The authors noted that there are few such studies in the field of dental hygiene. The study was conducted during one semester and aimed to evaluate the improvement of problem-solving ability and self efficacy. The intervention was used during a course designed to achieve integrated thinking skills.
It appears that the instrument used for the intervention was developed by the researchers. It appears that the instrument was used before and after the course. The instrument was tested for reliability using “Cronbach’s alpha.” It should be noted tha Doctor t Cronbach preferred to use the term rock coefficient alpha rather than Cronbach’s alpha. The reliability of the instrument tested as high at 0.92. The researchers tested for normality (suggesting a high level of statistical integrity) and used a paired samples t-test and Pearson correlation. In essence, the statistical design is simple and appropriate assuming that the data are distributed normally.
There are four main problems here that are not addressed. First, the authors have not conducted any validity analysis. The instrument may be reliable but we need to understand why it is valid. Second, while the instrument seems to have improved learning we would expect this from any educational intervention. The study design suggests that the intervention may have been appropriate but leaves open the question whether there might be other educational interventions that would be more appropriate. Because there is no comparison we are left wondering whether this intervention is superior to other educational approaches. Third, the authors do not describe how the students were selected. Were all students in the class part of the study? Did the students have any opportunity to opt out? Fourth, it would seem that the intervention occurred prior to the start of class and at the end of class. The intervention may have been effective in the short run but we are left with a lack of understanding regarding the long-term effectiveness of it.
The authors report differences in means overall and for specific items. For problem-solving ability scores improved significantly for 11 items did not improve significantly for 20 items. Moreover, effect sizes for a number of the items were small. Reporting an overall score which shows improvement may be misleading when so many items failed to achieve significance. Four self-efficacy the authors found an overall significant improvement. Here nine of the items achieved statistical significance and 15 did not. So the same observation applies for self-efficacy. It may be misleading to contend that the intervention improves self-efficacy when more than half of the items showed no significant difference.
Also problematic is the issue that the authors do not report the results of the test for normality. I assume that the data are normal or a Spearman correlation would’ve had to be done rather than a Pearson. Still, this needs to be reported in the paper.
The study finds a correlation coefficient of 0.701 significant at .01. I’m not quite sure what this really proves without some kind of more appropriate validity investigation. Could it be that the same researchers developed the questions for problem-solving capability and self-efficacy such that the correlation analysis only establishes that the same people develop the items. I’m also not sure what it means that there is a relationship between problem-solving capability and self-efficacy.
The authors note that the findings are similar to those of previous studies in dentistry, nursing and medicine. Our dental hygienists so different that a separately fashioned intervention and evaluation are necessary?
In terms of educational approach, I wonder whether problem solving ability should not be measured in clinical settings rather than using questionnaires. I would like to see some literature that compares didactic learning to clinical experience.
Further, I would be interested in the authors thoughts regarding how an intervention like this might to generalize to other similar programs within country and in other countries.
Author Response
Dear Reviewer 3.
Thank you very much for reviewing my paper. All your comments have contributed to making my paper of a higher quality. I have tried to make most of the changes to the part you suggested. I sincerely hope that the part I have revised will be agreeable with you.
This is a study of a problem-based learning intervention involving 31 fourth-ear dental hygienist students at Gangneung National University. The intervention consisted of 32 detailed items assessing problem-solving ability and 22 detailed questions relating to self-efficacy. The study found significant increases in both measures with an increase of 0.284 problem solving ability and 2.674 self-efficacy.
The research is straightforward and scientifically conducted. That notwithstanding, there are some pretty serious issues that need to be dealt with as described below.
Problem-based learning has been used in a wide variety of settings. The authors noted that there are few such studies in the field of dental hygiene. The study was conducted during one semester and aimed to evaluate the improvement of problem-solving ability and self efficacy. The intervention was used during a course designed to achieve integrated thinking skills.
It appears that the instrument used for the intervention was developed by the researchers. It appears that the instrument was used before and after the course. The instrument was tested for reliability using “Cronbach’s alpha.” It should be noted that Doctor t Cronbach preferred to use the term rock coefficient alpha rather than Cronbach’s alpha. The reliability of the instrument tested as high at 0.92. The researchers tested for normality (suggesting a high level of statistical integrity) and used a paired samples t-test and Pearson correlation. In essence, the statistical design is simple and appropriate assuming that the data are distributed normally.
- Thank you for your recommendation. I revised the term of Cronbach’s alpha to coefficient alpha in Materials and Methods section.
- Page 3. Line 123-125. 2. Materials and Methods (Red font)
Problem-solving ability comprised 32 items, whereas the mean reliability (coefficient α) for the questionnaire items was 0.88. A negative question was calculated using inverse calculation. Moreover, self-efficacy comprised 22 items with coefficient α of 0.92 for the questionnaire items.
-There are four main problems here that are not addressed. First, the authors have not conducted any validity analysis. The instrument may be reliable but we need to understand why it is valid.
- Thank you for your indication. We authors modified and used the instrument developed in the previous PBL studies through discussion among researchers to fit the purpose of this study. The instrument was developed and tested for validity in Heppner and Petersen’s study (1982) as below. Therefore, we authors thought the instrument was valid for evaluating the problem-solving ability among the students in this study.
- The reference: Heppner, P. P., & Petersen, C. H. (1982). The development and implications of a personal problem- solving inventory. Journal of Counseling Psychology, 29, 66-75.
-Second, while the instrument seems to have improved learning we would expect this from any educational intervention. The study design suggests that the intervention may have been appropriate but leaves open the question whether there might be other educational interventions that would be more appropriate. Because there is no comparison we are left wondering whether this intervention is superior to other educational approaches.
- Thank you for your opinion. It is possible for problem-solving ability and self-efficacy to be improved from other educational interventions as your comment. However, it has been reported in several previous studies that problem based learning (PBL) could be a good way to improve the skills such as problem-solving ability, self-directed learning, self-efficacy, etc. and we authors agreed the suggestions. In a step that we designed this study, although we considered the control and test group in order to compare between PBL and other educational intervention, we could not include the comparison because of operational limitations. However, it should be evaluated in further studies. We authors added the limitation and suggestion in Discussion section as below.
- Page 8. Line 212-215. 4. Discussion (Red font)
Because there were no comparisons between the PBL and other educational intervention, the effects of only PBL on the problem-solving ability and self-efficacy should be evaluated through the study design for controlling and adjusting other interventions in further studies.
-Third, the authors do not describe how the students were selected. Were all students in the class part of the study? Did the students have any opportunity to opt out?
- Thank you for your indication. We selected all students who participated in the class (Clinical Dental Hygiene for special patients). But, we suggested the informed consent for this study and informed sufficient opportunities to opt out. I added the description in the Materials and Methods section.
- Page 2. Line 87-88. 2. Materials and Methods (Red font)
We selected all students who participated in the class and suggested the informed consent for this study. We also informed sufficient opportunities to opt out.
-Fourth, it would seem that the intervention occurred prior to the start of class and at the end of class. The intervention may have been effective in the short run but we are left with a lack of understanding regarding the long-term effectiveness of it.
- Thank you for your comment. We authors agreed the short-term effectiveness by this study, not for the long-term effectiveness as your opinion. In order to evaluate the long-term effectiveness, we should design the longitudinal studies and evaluate the outcomes as time goes by. I described the suggestions in the Discussion section as your comment.
- Page 8. Line 219-220. 4. Discussion (Red font)
In this study, the main results mean the short-term effectiveness of PBL on problem-solving ability and self-efficacy. In order to evaluate the long-term effectiveness, we should design the longitudinal studies and evaluate the outcomes as time goes by.
-The authors report differences in means overall and for specific items. For problem-solving ability scores improved significantly for 11 items did not improve significantly for 20 items. Moreover, effect sizes for a number of the items were small. Reporting an overall score which shows improvement may be misleading when so many items failed to achieve significance. Four self-efficacy the authors found an overall significant improvement. Here nine of the items achieved statistical significance and 15 did not. So the same observation applies for self-efficacy. It may be misleading to contend that the intervention improves self-efficacy when more than half of the items showed no significant difference.
- Thank you for your indication. It may be misleading as the overall significant improvement when considering the overall scores in the problem-solving ability and self-efficacy improvement. Therefore, I described additionally the possibility of misleading and tried to make readers understand the results clearly in the Discussion section.
- Page 8. Line 205-209. 4. Discussion (Red font)
- One thing to note is for the levels of problem-solving ability and self-efficacy improved significantly for overall scores and some specific items did not be improved significantly. It may be misleading as the overall significant improvement when considering only the overall scores in the problem-solving ability and self-efficacy improvement.
Also problematic is the issue that the authors do not report the results of the test for normality. I assume that the data are normal or a Spearman correlation would’ve had to be done rather than a Pearson. Still, this needs to be reported in the paper.
- Thank you for your indication. The data were normal and ordinal scale. Spearman correlation has been done for re-analysis. And I corrected the description and the results as 0.626 in the Materials and Methods and Results section.( Page 7)
The study finds a correlation coefficient of 0.701 significant at .01. I’m not quite sure what this really proves without some kind of more appropriate validity investigation. Could it be that the same researchers developed the questions for problem-solving capability and self-efficacy such that the correlation analysis only establishes that the same people develop the items. I’m also not sure what it means that there is a relationship between problem-solving capability and self-efficacy.
- Thank you for your opinion. As mentioned above, we authors modified and used the instrument developed and tested for validity in the previous PBL studies through discussion among researchers to fit the purpose of this study. Therefore, we authors thought the instrument was valid for evaluating the problem-solving ability and self-efficacy among the students in this study.
-The authors note that the findings are similar to those of previous studies in dentistry, nursing and medicine. Our dental hygienists so different that a separately fashioned intervention and evaluation are necessary?
- Thank you for your comment. I think that educational intervention focusing on the cases of subjects with integrative problems is essential because dental hygienists are oral experts who have to solve the client’s health problems based on critical thinking and integrative thinking. A problem-based learning method could be a effective way to train the skills, and it is also the goal of this study.
-In terms of educational approach, I wonder whether problem solving ability should not be measured in clinical settings rather than using questionnaires. I would like to see some literature that compares didactic learning to clinical experience.
Further, I would be interested in the authors thoughts regarding how an intervention like this might to generalize to other similar programs within country and in other countries.
- Thank you for your opinion. We authors think these abilities should be trained and evaluated in clinical settings as well as educational settings. Until now, there were few trials to train the skills and to be assessed, we authors determined the order of priority between the clinical- and educational- settings. In further studies, the improvement in clinical settings should be assessed and the module for the training should be developed. In addition, we authors would like for this course to expand in other countries or other settings. In order to do this, we need to develop a standardized curriculum and textbooks to enable this kind of learning.

Round 2
Reviewer 2 Report
The authors did well to respond to the requested feedback. The revised manuscript is clearer and more detailed. One minor point:
1. Line 204-208 – This statement is unclear – “One thing to note is for the levels of problem-solving ability and self-efficacy improved significantly for overall scores and some specific items did not be improved significantly. It may be misleading as the overall significant improvement when considering only the overall scores in the problem-solving ability and self-efficacy improvement”. Check grammar in ‘did not be improved…’. Also, the second sentence in the statement is incomplete and unclear.
Otherwise, the manuscript is improved. Upon addressing this item, the manuscript appears suitable for publication.
Author Response
Thank you for your indication. I revised clearly the sentences as below:
One thing to note is that the level of problem-solving ability and self-efficacy were significantly improved in overall score, and some specific items were not significantly improved. Considering only the overall score of improvement in problem-solving ability and self-efficacy, it might be mistaken for a comprehensively significant improvement.

Reviewer 3 Report
No further comments. Concerns have been addressed
Author Response
Thank you for reviewing till the end.